# Correlating STED and synchrotron XRF nano-imaging unveils cosegregation of metals and cytoskeleton proteins in dendrites

Florelle Domart[1,2,3], Peter Cloetens[4], Stéphane Roudeau[1,2], Asuncion Carmona[1,2], Emeline Verdier[3], Daniel Choquet[3,5†], Richard Ortega[1,2†]*

[1]Chemical Imaging and Speciation, CENBG, Univ. Bordeaux, Gradignan, France; [2]CNRS, IN2P3, CENBG, UMR 5797, Gradignan, France; [3]Univ. Bordeaux, CNRS, Interdisciplinary Institute for Neuroscience, IINS, UMR 5297, Bordeaux, France; [4]ESRF, the European Synchrotron, Grenoble, France; [5]Univ. Bordeaux, CNRS, INSERM, Bordeaux Imaging Center, BIC, UMS, Bordeaux, France

**Abstract** Zinc and copper are involved in neuronal differentiation and synaptic plasticity but the molecular mechanisms behind these processes are still elusive due in part to the difficulty of imaging trace metals together with proteins at the synaptic level. We correlate stimulated-emission-depletion microscopy of proteins and synchrotron X-ray fluorescence imaging of trace metals, both performed with 40 nm spatial resolution, on primary rat hippocampal neurons. We reveal the co-localization at the nanoscale of zinc and tubulin in dendrites with a molecular ratio of about one zinc atom per tubulin-$\alpha\beta$ dimer. We observe the co-segregation of copper and F-actin within the nano-architecture of dendritic protrusions. In addition, zinc chelation causes a decrease in the expression of cytoskeleton proteins in dendrites and spines. Overall, these results indicate new functions for zinc and copper in the modulation of the cytoskeleton morphology in dendrites, a mechanism associated to neuronal plasticity and memory formation.

*For correspondence: ortega@cenbg.in2p3.fr

†These authors contributed equally to this work

Competing interests: The authors declare that no competing interests exist.

## Introduction

The neurobiology of copper and zinc is a matter of intense investigation since they have been recently associated to neuronal signaling and differentiation processes (*Chang, 2015*; *Vergnano et al., 2014*; *Barr and Burdette, 2017*; *D'Ambrosi and Rossi, 2015*; *Xiao et al., 2018*; *Hatori et al., 2016*). Zinc ions are stored in pre-synaptic vesicles in a subset of glutamatergic neurons. They are released in the synaptic cleft during neurotransmission. Zinc ions are effectors of synaptic plasticity by modulating the activity of neurotransmitter receptors such as NMDA (N-methyl-d-aspartate) and AMPA (α-amino-3-hydroxy-5-methyl-4-isoxazolepropionic acid) receptors (NMDARs and AMPARs) (*Vergnano et al., 2014*; *Barr and Burdette, 2017*). The role of copper in neural functions is less well described than that of zinc, but copper ions are also known to be released in the synaptic cleft where they can modulate the activity of neurotransmitter receptors (*D'Ambrosi and Rossi, 2015*). Copper is involved in broad neuronal functions such as the regulation of spontaneous neuronal activity or of rest–activity cycles (*Chang, 2015*; *Xiao et al., 2018*), and more generally in neuronal differentiation (*Hatori et al., 2016*). In addition to these functions of the labile fraction of metals on neuronal signaling and differentiation, we have recently indicated that copper and zinc could be involved in the structure of dendrites and dendritic spines (*Perrin et al., 2017*). We have shown on cultured hippocampal neurons the localization of zinc all along dendritic shafts and that of copper in the neck of dendritic spines, suggesting their interaction with cytoskeleton proteins.

Understanding the functions of copper and zinc in the cytoskeleton structure would require however to correlate metal localization with respect to relevant cytoskeleton proteins at the sub-dendritic level. To this end, we have developed an original method for correlative imaging of metals and proteins at 40 nm spatial resolution described in this paper.

Nano-Synchrotron X-Ray Fluorescence (Nano-SXRF) imaging using hard X-ray beams, typically above 10 keV energy, is a powerful technique to investigate the cellular localization of metals since it allows the mapping of element distributions in single cells with high analytical sensitivity (*Victor et al., 2018*). Using Kirkpatrick–Baez (KB) focusing mirrors, SXRF has reached a spatial resolution of 13 nm on ID16A beamline at the European Synchrotron Radiation Facility (ESRF), while maintaining a high photon flux as required for detecting trace elements (*da Silva et al., 2017*). We have previously reported a correlative microscopy approach consisting in labeling organelles or proteins with specific fluorophores for live-cell imaging prior to SXRF imaging (*Roudeau et al., 2014*; *Carmona et al., 2019*). This correlative approach is limited by the spatial resolution of optical fluorescence microscopy, above 200 nm, which is larger than the spatial resolution achieved today with nano-SXRF and insufficient to resolve synaptic sub-structures. To overcome this limitation we present a method to correlate nano-SXRF with STED (STimulated-Emission-Depletion microscopy) performed both at 40 nm resolution. With the combination of these two high-resolution imaging techniques we observe trace metals co-localization with cytoskeleton proteins at the synaptic level in rat hippocampal neurons. Furthermore, we explored the effect of zinc deficiency induced by TPEN, an intracellular zinc-chelator, on the expression of cytoskeleton proteins in dendrites and dendritic spines.

## Results

### Super-resolution live-cell STED microscopy and nano-SXRF imaging combination

We designed a specific protocol consisting in live-cell STED microscopy on silicon nitride (SN) substrates followed by cryogenic processing of the cells before nano-SXRF imaging as described in (*Figure 1*). Primary rat hippocampal neurons were cultured in vitro during 15 days (DIV15) on sterile SN membranes placed above an astrocyte feeder layer as adapted from *Kaech and Banker, 2006*; *Perrin et al., 2015*. SN membranes with an orientation frame were used to facilitate the reproducible positioning of the samples during the correlative imaging procedure (*Figure 1a*). SN membranes are biocompatible, trace metal free, 500 nm thin, transparent, flat and rigid supports developed for X-ray microscopy. To perform STED microscopy we stained the DIV15 primary rat hippocampal neurons with either SiR-actin, SiR-tubulin, or with SiR-tubulin and SiR700-actin together. These two far-red SiR-based fluorophores have been developed for live-cell super-resolution microscopy and are based respectively on the actin ligand jasplakinolide and the tubulin ligand docetaxel (*Lukinavičius et al., 2014*; *Lukinavičius et al., 2016*). Cytoskeletal SiR-based fluorescent dyes have several features that were important for the success of this correlative imaging approach. On the one hand, they are markers suitable for STED imaging that do not require overexpression of the proteins. On the other hand, when these molecules bind to proteins of the cytoskeleton they block the cell structure preventing movement of the regions of interest in the time span between living cells STED imaging and cryofixation. Confocal and STED live-cell images were obtained using a commercial Leica DMI6000 TCS SP8 X with a 93x objective at immersion in glycerol and numerical aperture of 1.3 (*Figure 1b*). STED images were recorded at 40 nm spatial resolution. Then the SN membranes were plunge-frozen at −165°C and freeze-dried at −90°C under secondary vacuum before analysis on ID16A beamline at ESRF (*Figure 1b*). Nano-SXRF and synchrotron X-ray phase contrast imaging (PCI) were performed with an X-ray beam size of 40 nm at 17 KeV energy (*Figure 1b*). Nano-SXRF, PCI, and STED microscopy images can be merged to perform multimodal correlative microscopy (*Figure 1c*) of chemical elements, electron density, and fluorescently labeled proteins all obtained at similar spatial resolution (<40 nm). Post-acquisition alignment of the multimodal images was performed using intra-specimen localization structures such as characteristic filament crosses, or dendritic protrusions. The result is accurate despite the freeze-drying process that can marginally alter the cell structure (*Figure 1c*). The adequacy of the sample preparation method and of the post-acquisition alignment of the images for the correlative purpose was checked by comparing live-cell STED with STXM (Scanning Transmission X-ray Microscopy) of freeze-dried cells obtained with 25

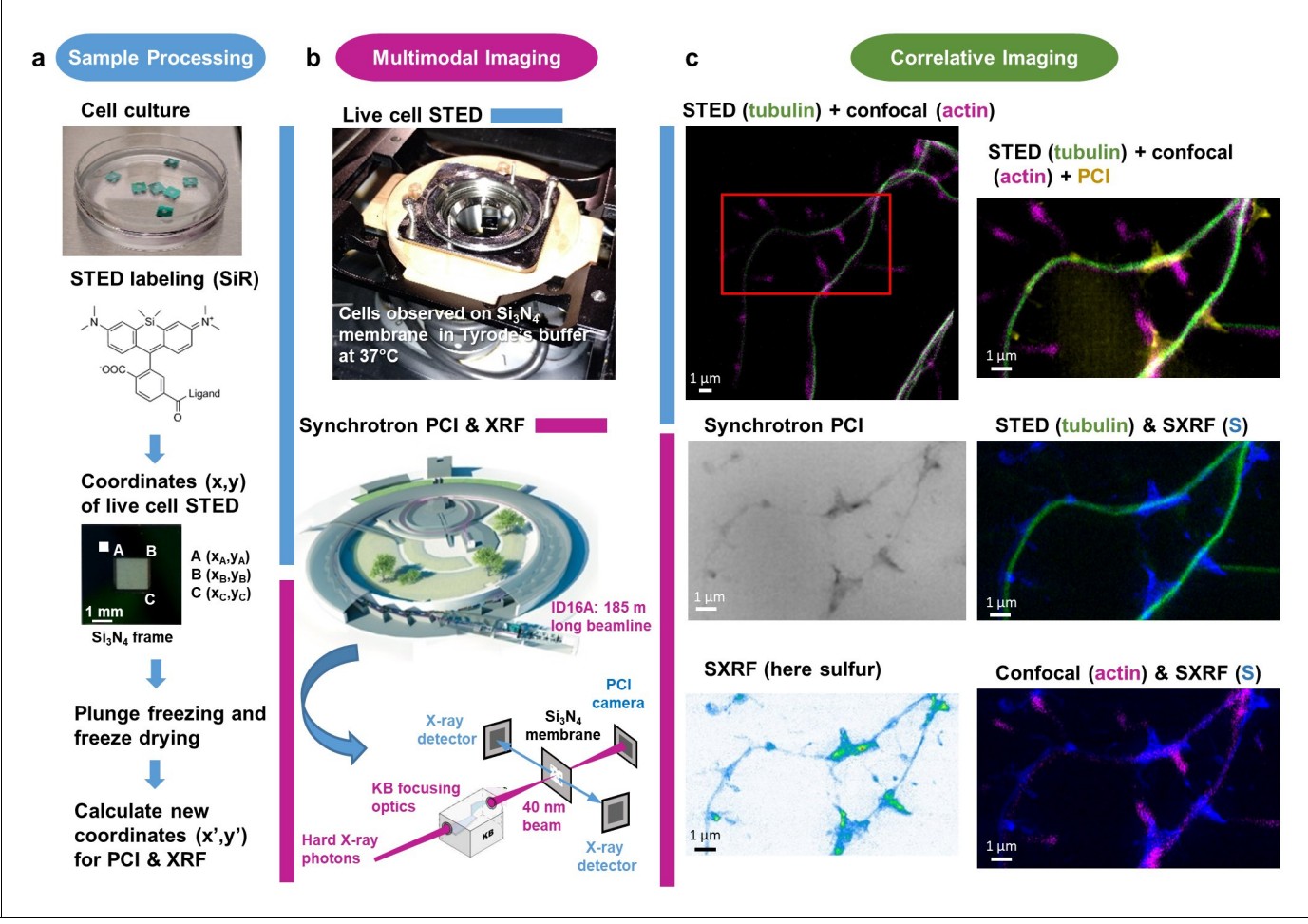

**Figure 1.** Workflow for correlative STED and synchrotron multimodal nano-imaging. (**a**) Sample processing. Primary neurons are cultured on silicon nitride membranes and labeled with fluorescent probes designed for STED microscopy such as SiR-tubulin or SiR700-actin. STED microscopy is performed on living cells and orthonormal coordinates (x,y) of regions of interest are recorded. Immediately after STED microscopy cells are plunge-frozen and freeze-dried. New coordinates (x',y') of the regions of interest are calculated to perform XRF and PCI imaging on the synchrotron microscope. (**b**) Multi-modal imaging. Live-cell STED and confocal microscopy are performed within a thermalized chamber. Synchrotron XRF and PCI are carried out on freeze-dried samples. The KB optics are 185 m away from the X-ray source enabling to focus hard X-rays at 40 nm beam size. (**c**) Correlative imaging. Overlay images of STED, confocal, synchrotron PCI and XRF are produced on areas of few tens of μm large with a spatial resolution of 40 nm for STED, 30 nm for PCI and 40 nm for SXRF. Several elemental maps (here sulfur) can be super-imposed with protein distributions (i.e. actin or tubulin) in dendrites and spines at 40 nm spatial resolution.

The online version of this article includes the following figure supplement(s) for figure 1:

**Figure supplement 1.** Correlation of STED imaging on live cells and of STXM microscopy after cryofixation and freeze-drying.

nm spatial resolution at HERMES beamline, SOLEIL synchrotron. STXM enabled to visualize the dendritic morphology showing the exact superposition of dendrites after cryofixation and freeze- drying with SiR-tubulin fluorescence previously observed by STED on living cells (*Figure 1—figure supplement 1*). The limit of detection (LOD) of nano-SXRF analysis was calculated for the detected elements according to IUPAC (International Union of Pure and Applied Chemistry) guideline resulting for zinc in in a 0.009 ng.mm$^{-2}$ LOD, corresponding to 14 zeptogram of zinc (about 130 atoms) per pixel of 40 nm x 40 nm size (*Table 1*).

## Zinc is highly concentrated in dendritic spines

The highest zinc content is found in F-actin-rich dendritic spines where zinc is highly correlated with sulfur (Person's correlation coefficient >0.6) (*Figures 2–4* and *Figure 2—figure supplement 1*; *Figure 3—figure supplement 1*; *Figure 4—figure supplement 1*). Zinc distribution however is not fully

**Table 1.** LOD calculated according to IUPAC (International Union of Pure and Applied Chemistry) guideline as derived from 12 blank measurements for phosphorus, sulfur, potassium and zinc, expressed in ng.mm$^{-2}$, and expressed in g (and number of atoms) within pixels of 40 nm x 40 nm size.

| | Phosphorus | Sulfur | Potassium | Zinc |
|---|---|---|---|---|
| LOD | 0.513 ng.mm$^{-2}$ | 0.138 ng.mm$^{-2}$ | 0.152 ng.mm$^{-2}$ | 0.009 ng.mm$^{-2}$ |
| LOD in 40 nm x 40 nm pixel | 8.2 10$^{-19}$ g (16,000 atoms) | 2.2 10$^{-19}$ g (4200 atoms) | 2.4 10$^{-19}$ g (3800 atoms) | 1.4 10$^{-20}$ g (130 atoms) |

The online version of this article includes the following source data for Table 1:

**Source data 1.** LOD calculation data.

superimposed with F-actin localization suggesting that both entities are not directly in interaction in dendritic protrusions. Zinc and sulfur hot spots are found within narrow regions in the dendritic spines characterized by the highest electron densities as shown by synchrotron PCI maps (*Figure 3* and *Figure 4—figure supplement 1*). Although in lower concentration than zinc, copper was also detected in F-actin-rich dendritic spines showing a distinct distribution pattern than for Zn (*Figure 3* and *Figure 3—figure supplement 1*). Copper is mainly located at the basis of the dendritic protrusions (*Figure 3h*) where it is co-localized with the highest fluorescence signal of F-actin (*Figure 3j*). The co-localization of copper and F-actin in dendritic protrusions is only partial since copper may not be detected in regions of weaker SiR-actin fluorescence. In rare cases iron was also detected, as 200 nm x 100 nm structures of locally dense iron spots, in the dendrite, contiguous to the dendritic spines (*Figure 4—figure supplement 1K*).

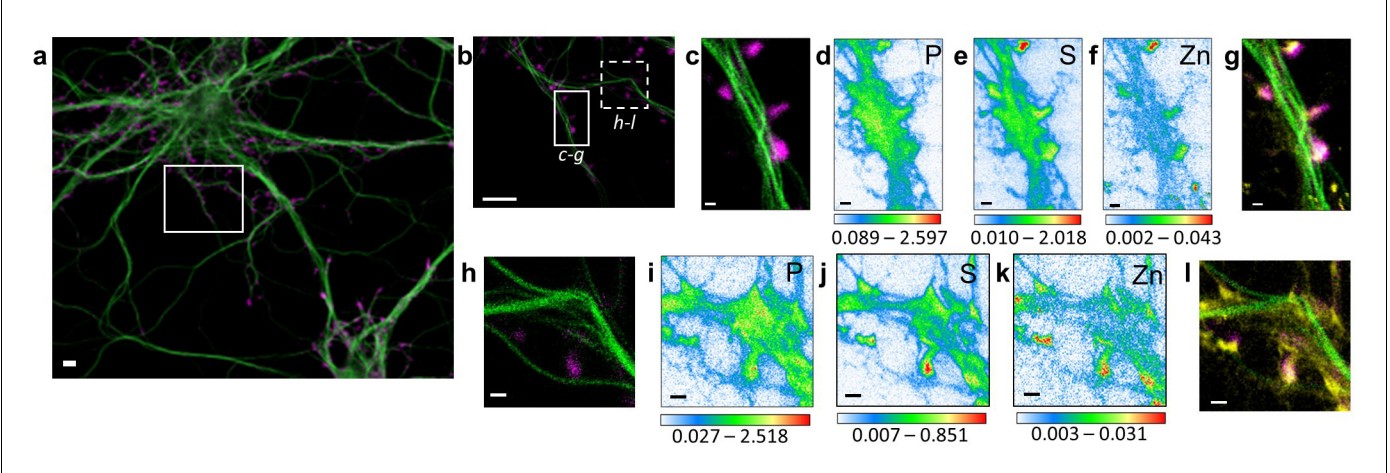

**Figure 2.** Correlative STED nano-SXRF element imaging in dendritic spines. (a) Confocal image from a DIV15 primary rat hippocampal neuron stained with SiR-tubulin (green) and SiR700-actin (magenta). (b) STED image of SiR-tubulin (green) and confocal SiR700-actin (magenta) from the framed region shown in (a). (c) STED image of SiR-tubulin (green) and confocal SiR700-actin (magenta) for the region mapped by SXRF shown in (b) in plain line. (d) Phosphorus SXRF map. (e) Sulfur SXRF map. (f) Zinc SXRF map. (g) Merged images of zinc SXRF map (yellow), STED SiR-tubulin (green) and confocal SiR700-actin (magenta). (h) STED image of SiR-tubulin (green) and confocal SiR700-actin (magenta) image from the framed region shown in (b) in dotted line. (i) Phosphorous SXRF map. (j) Sulfur SXRF map. (k) Zinc SXRF map. (l) Merged images of zinc SXRF map (yellow), STED SiR-tubulin (green), and confocal SiR700-actin (magenta). Scale bars: 500 nm, except (a) and (b) 5 μm. Color scale bars: min-max values in ng.mm$^{-2}$.

The online version of this article includes the following source data and figure supplement(s) for figure 2:

**Figure supplement 1.** Element distributions in dendritic spines.

**Figure supplement 1—source data 1.** Data for Pearson's correlation coefficients of *Figure 2—figure supplement 1* panel h.

**Figure supplement 1—source data 2.** Data for Pearson's correlation coefficients of *Figure 2—figure supplement 1* panel o.

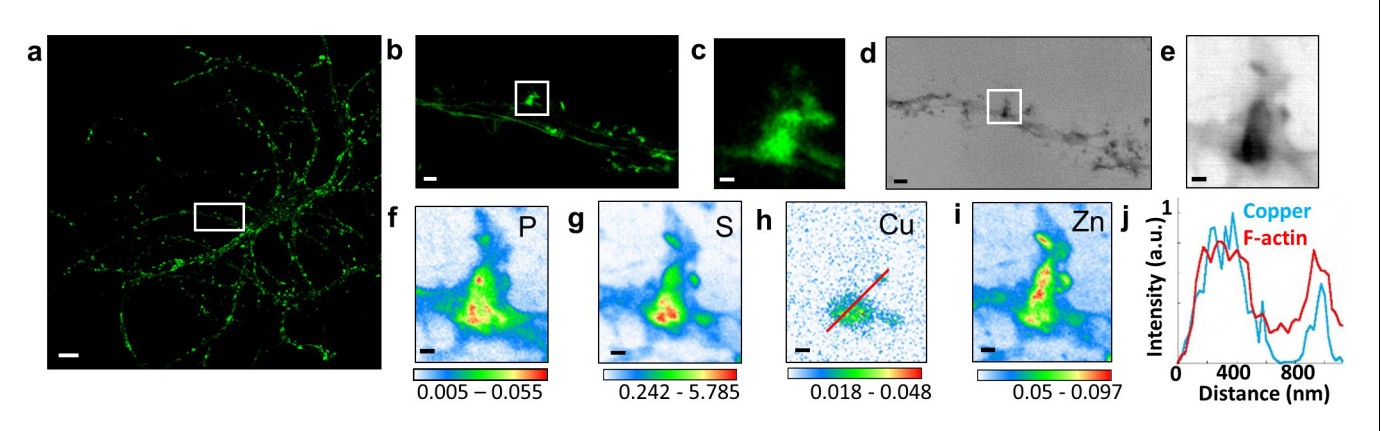

**Figure 3.** Correlative STED microscopy of SiR-actin with synchrotron PCI and XRF in spines. (**a**) Confocal imaging of a primary rat hippocampal neuron labeled with SiR-actin showing F-actin-rich protrusions along dendrites. (**b**) STED imaging of SiR-actin of the dendrite region framed in white in a. (**c**) Zoom on the STED SiR-actin of a F-actin-rich protrusion framed in (b). (**d**) Synchrotron radiation X-ray PCI of the dendrite region framed in white in a. (**e**) Zoom on the PCI region of a F-actin-rich protrusion framed in (d). (**f-i**) SXRF element maps (P, S, Cu, Zn) from the region of interest framed in (b) and (d). (**j**) Line scans for F-actin (red) and copper (blue) normalized distributions along the red line plotted in (h). Scale bar: 200 nm, except for (a) 10 μm, (b) and (d) 1 μm. Color scales: min-max values in ng.mm$^{-2}$.

The online version of this article includes the following figure supplement(s) for figure 3:

**Figure supplement 1.** Element distributions in dendrites and spines.

## Zinc and microtubules are co-localized

The most striking result of this correlative microscopy approach is the observation of zinc and tubulin co-localization in thin dendritic processes as illustrated in *Figures 4–5* and *Figure 3—figure supplement 1*, *Figure 4—figure supplement 1*, *Figure 5—figure supplement 1*. In *Figure 5*, a region of interest (roi) showing parallel microtubule filaments observed by STED microscopy was selected (*Figure 5a*). These parallel tubulin filaments could not be resolved by confocal microscopy (*Figure 5b*), but only by STED microscopy (*Figure 5c*). Similarly to STED microscopy, nano-SXRF performed on the same roi was able to separate the element distributions of the two thin dendritic processes (*Figure 5d–e*). Plot profiles of zinc and tubulin relative signal intensities across the two tubulin filaments shows the co-localization of their distributions (*Figure 5f*). The very good superimposition of S and Zn distributions along tubulin imaged by STED is systematically observed (*Figures 4* and *5k* and *Figure 3—figure supplement 1*, *Figure 4—figure supplement 1*, *Figure 5—figure supplement 1*). For thinner dendrites, zinc is superimposed with the dendritic structure, while for thicker branches zinc is located in sub-dendritic regions (*Figure 5—figure supplement 1*). The quantitative analysis of nano-SXRF data for 21 regions showing Zn and tubulin co-localization indicates that the atomic ratio S/Zn is of 43 ± 11 (*Figure 5l* and *Supplementary file 1*). Considering that the amino acid sequence of the rat tubulin-α/β dimer contains 49 sulfur atoms per dimer, from methionine and cysteine residues, the S/Zn ratio in microtubules corresponds to a theoretical tubulin-α/β dimer over Zn ratio of 0.9 ± 0.2 (*Figure 5l*).

## Zinc depletion disturbs the dendritic cytoskeleton

Primary rat hippocampal neurons cultured on glass coverslips were exposed to TPEN (N,N,N′,N′-tetrakis(2-pyridylmethyl)ethylenediamine), a zinc intracellular chelator. At DIV 15, mature neurons were exposed to a subcytotoxic concentration of 5 μM TPEN during 24 hr, or simultaneously to 5 μM TPEN and 10 μM ZnCl$_2$, or to 0.1% ethanol (v/v) for the control group. Subcytotoxic concentration of TPEN during 24 hr exposure time was selected to study the effect of mild zinc depletion conditions on neuronal differentiation, avoiding cytoskeleton alteration that would result from direct cytotoxic effects at higher TPEN concentrations. Quantitative fluorescence microscopy of β-tubulin and F-actin expression was assessed in dendritic branches. Representative examples of F-actin and β-tubulin labeling in dendrites for each of the three experimental groups are shown in *Figure 6a–c*. Fluorescence intensities of F-actin and β-tubulin were quantified in dendritic branches from more

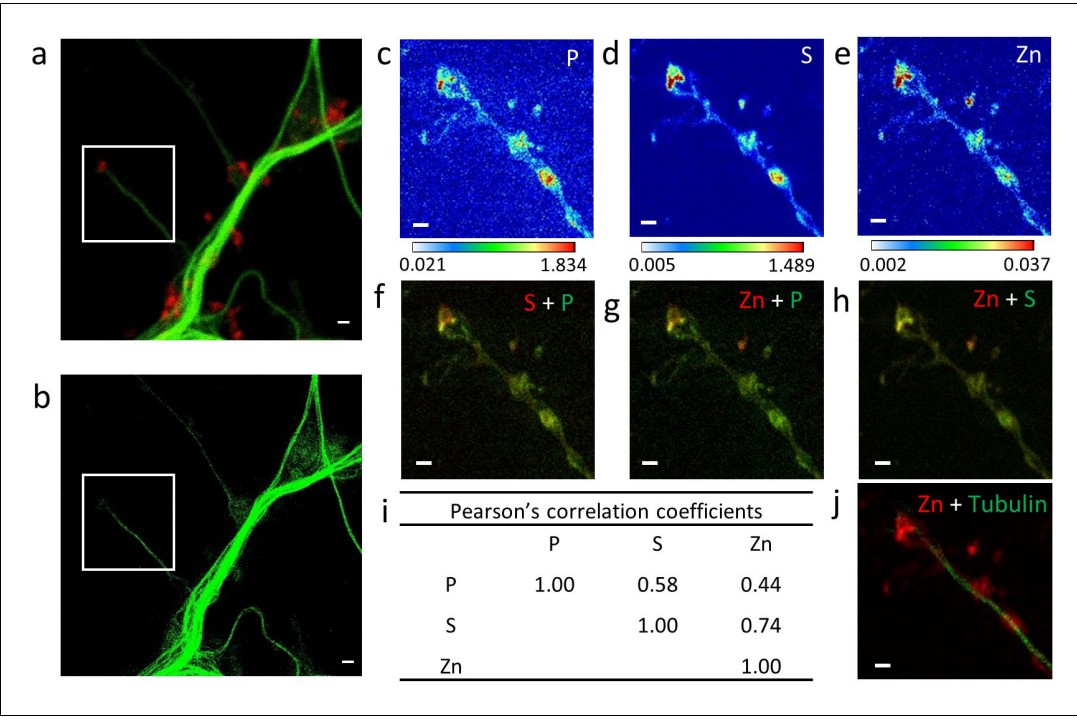

**Figure 4.** Correlative imaging in dendrites and spines. (**a**) Confocal image of SiR-tubulin (green) and SiR700-actin (red). (**b**) STED image of SiR-tubulin (green) for the same zone as in (a). (**c-e**) SXRF element maps (P, S, Zn) from the region of interest framed in (a) and (b). (**f**) Overlay image of phosphorus (green) and sufur (red). (**g**) Overlay image of phosphorus (green) and zinc (red). (**h**) Overlay image of sulfur (green) and zinc (red). (**i**) Pearson's correlation coefficients for the elements (P, S, Zn) in the roi. (**j**) Overlay image of STED SiR-tubulin (green) and zinc (red). Scale bars: 500 nm, except for (a) and (b) 1 µm. Color scales min-max values in ng.mm$^{-2}$.

The online version of this article includes the following source data and figure supplement(s) for figure 4:

**Source data 1.** Data for Pearson's correlation coefficients.
**Figure supplement 1.** Multimodal imaging of dendrites and spines.

than 60 different neurons and were normalized with respect to the median value of the control group (*Figure 6d–e*). The experiment was repeated on three biological replicates (*Figure 6—figure supplement 1*). Zinc chelation with 5 µM TPEN results in a statistically significant decrease of β-tubulin and F-actin fluorescence compared to the control group (*Figure 6d–e* and *Figure 6—figure supplement 1*). The decrease in β-tubulin and F-actin expression following TPEN exposure is reversed by the addition of Zn strongly suggesting that the effect of TPEN is due to Zn chelation and not to unspecific interactions (*Figure 6d–e* and *Figure 6—figure supplement 1*).

## Discussion

Due to their low concentration in cells, the investigation of zinc and copper functions requires sensitive analytical methods and, in particular, the identification of the proteins interacting with those metals remains a difficult analytical challenge. In this context, we combined two high-resolution chemical imaging methods, STED super-resolution photonic fluorescence microscopy (*Hell and Wichmann, 1994*; *Nagerl et al., 2008*) and synchrotron XRF nano-imaging (*Victor et al., 2018*) to gather information on protein and element distributions at the nanometer scale in cells. STED microscopy has already been successfully combined to transmission electron microscopy to correlate protein localization with cell ultrastructure (*Watanabe et al., 2011*), or to atomic force microscopy to investigate protein aggregation at high spatial resolution (*Cosentino et al., 2019*). Recently, STED has been performed together with synchrotron scanning diffraction microscopy to inform about diffraction patterns in cells (*Bernhardt et al., 2018*). These correlative approaches however cannot be transposed to the imaging of metals in cells since they require steps of chemical fixation

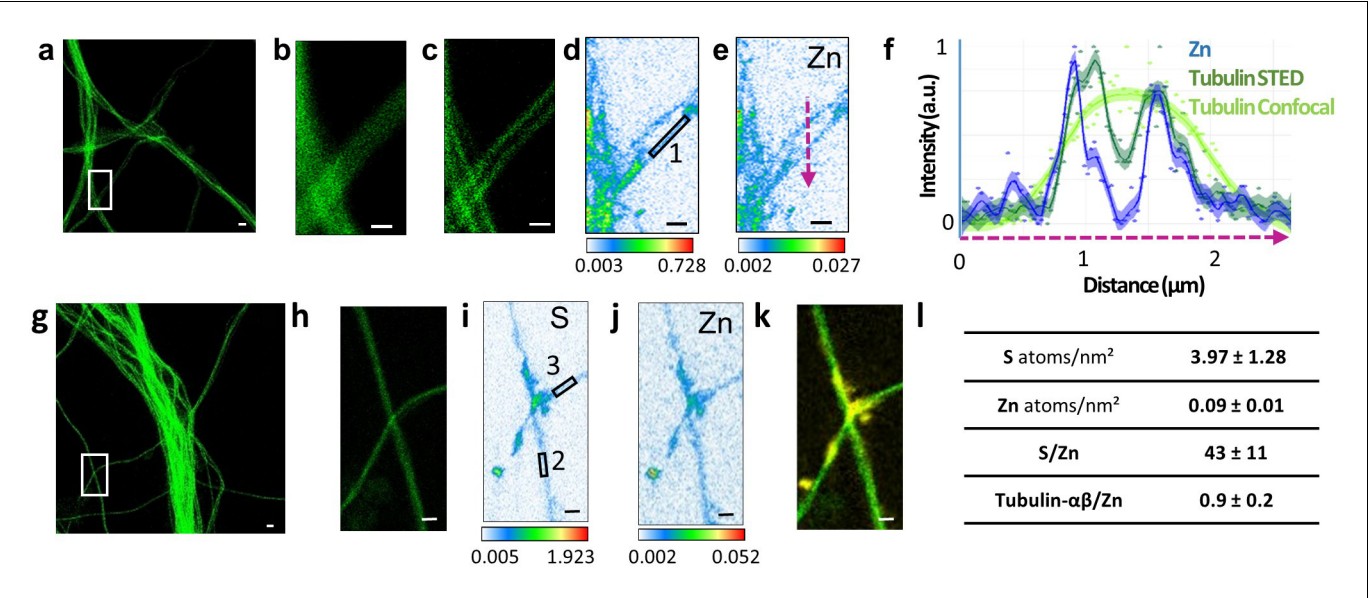

**Figure 5.** Correlative STED and nano-SXRF imaging in dendrites. (a) STED image of SiR-tubulin (green). (b) Confocal SiR-tubulin image of the region framed in (a). (c) STED SiR-tubulin image of the region framed in (a). (d) Sulfur SXRF map. (e) Zinc SXRF map. (f) Intensity plot profiles of zinc (blue), STED SiR-tubulin (dark green), and confocal SiR-tubulin (light green) along a line scan crossing two thin dendrites (magenta dotted line in (e)). (g) STED image of SiR-tubulin (green) in dendrites. (h) STED SiR-tubulin of the region framed in (g). (i) Sulfur SXRF map. (j) Zinc SXRF map. (k) Overlay image of STED SiR-tubulin (green) and zinc distribution (yellow). (l) Quantitative data analysis of the number of sulfur and zinc atoms.nm$^{-2}$ for 21 regions of interest centered on thin microtubules as illustrated for roi 1 to 3 framed in D and I (mean ± SD; n = 21; see also **Supplementary file 1**). Scale bars: 500 nm, except for (a) and (g) 1 µm.

The online version of this article includes the following figure supplement(s) for figure 5:

**Figure supplement 1.** Multimodal imaging of dendrites.

known to disrupt the metal-binding equilibrium in cells (**Roudeau et al., 2014**; **Perrin et al., 2015**). Moreover, glass coverslips used for STED microscopy usually contain significant amounts of zinc and of some other trace metals. We designed an original protocol to perform SXRF and STED correlative microscopy that fulfils specific requirements in terms of substrate for cell culture and protocols for sample preparation to avoid element contamination, loss or redistribution. Our protocol combines high-resolution imaging (40 nm) and high elemental sensitivity (zeptogram level). We focused our investigations on metal interactions with two cytoskeleton proteins, F-actin and tubulin based on our previous SXRF nano-imaging results showing the localization of zinc in dendritic shafts and of copper in the spines of hippocampal neurons (**Perrin et al., 2017**).

Zinc and copper are essential metals for neuronal functions (**Chang, 2015**; **Vergnano et al., 2014**; **Barr and Burdette, 2017**; **D'Ambrosi and Rossi, 2015**; **Xiao et al., 2018**; **Hatori et al., 2016**). Zinc is present in different regions of the brain, mainly the hippocampus and the amygdala (**Frederickson et al., 2000**). Zinc is largely complexed to proteins and is required for the functioning of hundreds of proteins. Only 10% of the total zinc is in a labile state, often referred as zinc ions ($Zn^{2+}$), a more easily exchangeable and therefore mobile element pool (**Takeda, 2001**). In a subset of glutamatergic neurons, this labile zinc is highly concentrated at the pre-synaptic level where zinc is stored in synaptic vesicles (**Vergnano et al., 2014**; **Barr and Burdette, 2017**). As for other neuro-transmitters, zinc is released in the synaptic cleft where it inhibits neuronal transmission mediated by AMPARs or NMDARs (**Vergnano et al., 2014**; **Barr and Burdette, 2017**; **Kalappa et al., 2015**). Zinc dyshomeostasis is broadly associated to brain diseases such as age-related cognitive decline, depression, or Alzheimer's disease (**Portbury and Adlard, 2017**).

The highest zinc content is found in dendritic spines, mainly located at the head of the spines, and zinc distribution was correlated with the higher cellular density as shown by synchrotron PCI. This result is in agreement with the expected location of zinc in the postsynaptic density (PSD) of

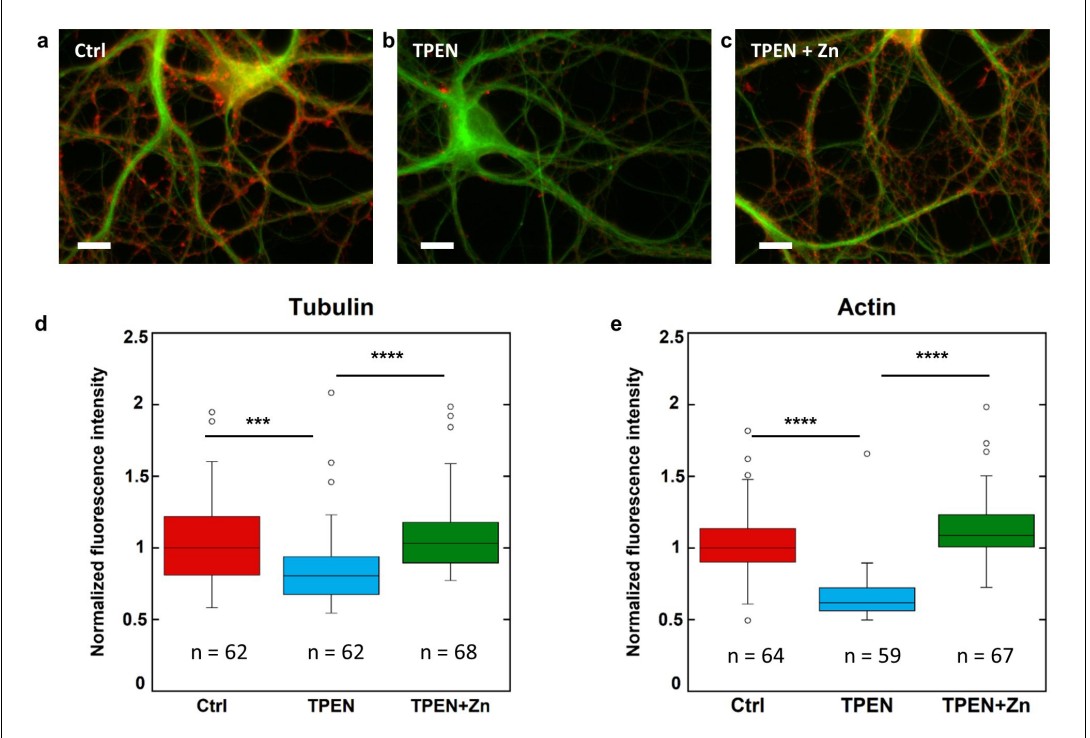

**Figure 6.** β-tubulin and F-actin fluorescence in control, TPEN, or TPEN and Zn treated neurons. (**a**) Representative fluorescence microscopy images of F-actin (red) and β-tubulin (green) in dendrites from control neurons. Scale bar = 10 µm. (**b**) TPEN treated neurons (5 µM, 24 hr). (**c**) TPEN (5 µM, 24 hr) and Zn (10 µM, 24 hr) treated neurons. (**d**) Comparison of β-tubulin normalized fluorescence intensities. (**e**) Comparison of F-actin normalized fluorescence intensities. Significant Kruskal-Wallis test (p<0.01) was followed by Dunn's test with p-values adjusted for pairwise comparisons: ***adj.p-value<0.001, ****adj.p-value<0.0001 (detailed in ***Supplementary file 2***).

The online version of this article includes the following source data and figure supplement(s) for figure 6:

**Source data 1.** Data for F-actin are available in file.
**Source data 2.** Data for β-tubulin are available in file.
**Figure supplement 1.** F-actin and β-tubulin fluorescence in control, TPEN, or TPEN and Zn treated neurons.
**Figure supplement 1—source data 1.** F-actin data for ***Figure 6—figure supplement 1*** are available in file.
**Figure supplement 1—source data 2.** Tubulin data for ***Figure 6—figure supplement 1*** are available in file.

hippocampal synapses where zinc could play a structural function in stabilizing proteins associated to the PSD such as Shank3 (***Tao-Cheng et al., 2016***).

We also observed the co-localization of zinc and tubulin in dendritic shafts at the nanometric level, including within the thinner dendrites. Based on the known amino acid composition of neuronal rat tubulin-αβ dimer, zinc co-localization with microtubules results in a stoichiometry of 0.9 ± 0.2 zinc atom per tubulin-αβ dimer. This result is in excellent agreement with data obtained by the Scatchard method on purified microtubule preparations from pig brains showing that zinc could bind tubulin with a molecular ratio of 0.86 ± 0.18 zinc atom per tubulin-αβ dimer (***Hesketh, 1983***). The Scatchard method consists in measuring the affinity of tubulin for zinc ions by adding known quantities of zinc to the purified protein. Our result is also in agreement with X-ray crystallography data of the tubulin-α/β dimer predicting the presence of one putative zinc ion per dimer, located in the tubulin-α subunit of zinc-induced tubulin sheets (***Löwe et al., 2001***). Moreover, molecular modeling have identified six putative binding sites of zinc to tubulin, suggesting that zinc would stabilize the structure of microtubules by enhancing electrostatic interactions (***Craddock et al., 2012***). Although our imaging approach cannot prove a direct interaction between zinc and tubulin, our data were obtained directly on primary cultured neurons in physiological conditions, without zinc addition, supporting the hypothesis of a structural requirement for zinc in tubulin network assembly.

It has been hypothesized that zinc dyshomeostasis could alter tubulin dynamics (***Craddock et al., 2012***). In our study, this hypothesis is supported by the reduction of the neuronal cytoskeleton

induced by zinc depletion using the intracellular zinc-chelator TPEN. We found that TPEN exposure resulted in decreased tubulin expression and that this effect was reversed by Zn. The decrease in tubulin expression can be explained by a limited neuronal differentiation resulting in a reduced density of tubulin filaments. We also observe a concomitant decrease of F-actin expression that could be explained as the consequence of tubulin decrease since microtubules are involved in the regulation of dendritic spines through actin remodeling (*Jaworski et al., 2009*). These zinc chelation results are in agreement with our previous observation that zinc supplementation in the culture medium induces the increase of β-tubulin and F-actin expression in mature primary rat hippocampal neurons (*Perrin et al., 2017*).

Copper is an essential metal for various cellular functions such as respiration, defense against free radicals and neurotransmitter synthesis (*Chang, 2015*; *D'Ambrosi and Rossi, 2015*; *Xiao et al., 2018*; *Hatori et al., 2016*). Contrary to zinc ions, copper is not localized within synaptic vesicles but copper ions can be released into the synaptic cleft by means of copper transporter proteins (*D'Ambrosi and Rossi, 2015*). Copper can play a biphasic role on neural transmission mediated by AMPARs, either by blocking or increasing the neurotransmission depending on its concentration and time of exposure (*Peters et al., 2011*). As for zinc, a disturbed copper homeostasis is found in many neurological diseases (*D'Ambrosi and Rossi, 2015*). Copper deficiency is observed in brain regions targeted for neurodegeneration such as the substantia nigra in Parkinson's disease (*Davies et al., 2014*), or the hippocampus in Alzheimer's disease (*Xu et al., 2017*).

Our data highlight the specific localization of copper within F-actin-rich regions along dendrites. The position, size and shape of the F-actin protrusions observed by STED microscopy (*Figure 2* and *Figure 1—figure supplement 1*, *Figure 2—figure supplement 1*, *Figure 3—figure supplement 1*) are representative of dendritic spines (*Chazeau and Giannone, 2016*; *Sala and Segal, 2014*). Although actin-rich structures should predominantly be components of dendritic spines, other actin-rich structures may be present such as actin patches (*Konietzny et al., 2017*). Both dendritic spines and actin patches are important triggers of synaptic differentiation. It is noteworthy that copper distribution is only partially correlated with F-actin since copper is not detected in regions of lower SiR-actin fluorescence (*Figure 3*). This observation could be explained by two different mechanisms. Copper might bind directly to F-actin but copper content is below the detection limit in regions containing lower amount of F-actin. Copper might not bind directly to F-actin but rather to other potential Cu-binding biomolecules that can interact with F-actin in specific areas of the dendritic protrusions where F-actin content is high. In an IMAC study focused on the copper proteome of HepG2 cells, 48 cytosolic proteins and 19 microsomal proteins displayed Cu-binding ability, among them γ-actin was identified both in the microsomal and the cytosolic fractions (*Smith et al., 2004*). Since γ-actin is also highly expressed in neurons, this IMAC result could suggest a direct binding of copper to F-actin in hippocampal neurons as well. In this same study, cofilin was also identified as putative Cu-binding protein in the cytosolic fraction (*Smith et al., 2004*). Copper could interact indirectly with molecules involved in the regulation of F-actin network in the dendritic spine such as cofilin, explaining the partial co-localization with F-actin.

It is interesting to underline that iron was not detected within dendritic spines, meaning that iron content in this compartment is below the detection limit of the method, suggesting that contrary to copper and zinc, iron is probably not directly involved in the morphogenesis of the dendritic spines. On the other hand, iron was observed as local hot spots of few hundred of nanometers along dendrites, the exact nature of these local iron-enrichment remains to be elucidated. A possible explanation could be the presence of iron in mitochondria known to be present in dendrites of hippocampal neurons (*Bastian et al., 2019*).

From a methodological perspective, the combination of STED super-resolution microscopy and nano-SXRF imaging stands as a solid new tool for the identification of metalloproteins directly in cells by correlating cellular imaging methods at a supramolecular scale. This correlative imaging approach revealed the co-segregation of zinc and copper with respectively microtubules and F-actin. Further experiments are now required to elucidate the interaction mechanisms between these metals and the cytoskeleton architecture. Finally, given the importance of cytoskeleton proteins in the morphological plasticity of neuronal connections (*Sala and Segal, 2014*), our results may contribute to explain why copper and zinc are present in higher concentrations in the nervous system than in other organs.

# Materials and methods

**Key resources table**

| Reagent type (species) or resource | Designation | Source or reference | Identifiers | Additional information |
|---|---|---|---|---|
| Biological sample (*Rattus norvegicus*) | Primary rat hippocampal neurons | Janvier labs, St Berthevin, France | | Freshly isolated from E18 Sprague-Dawley rats |
| Antibody | Mouse IgG1 anti-beta tubulin (mouse, monoclonal) | Sigma Aldrich | Sigma T4026 RRID:AB_477577 | (1:3000) |
| Antibody | Goat anti Mouse IgG1 CF568 (goat, polyclonal) | Ozyme | Ozyme BTM20248 RRID:AB_10854985 | (1:500) |
| Chemical compound, drug | Phalloidin-Alexa647 | Thermo Fisher Invitrogen | A22287 | (1:40) |
| Chemical compound, drug | SiR-actin | Spirochrome | SC001 | 1 μM |
| Chemical compound, drug | SiR-tubulin | Spirochrome | SC002 | 1 μM |
| Chemical compound, drug | SiR700-actin | Spirochrome | SC013 | 1 μM |
| Chemical compound, drug | TPEN (N,N,N,N-Tetrakis(2-pyridylmethyl)ethylenediamine) | Sigma Aldrich | P4413 | 5 μM |
| Chemical compound, drug | Zn (zinc chloride) | Sigma Aldrich | Z0152 | 10 μM |
| Chemical compound, drug | cytosine- arabinofuranoside | Sigma Aldrich | C6645 | |
| Chemical compound, drug | Paraformaldehyde | EMS | EMS 15710 | Used at 2 or 4% |
| Chemical compound, drug | BrainPhys medium | STEMCELL | 05790 | |
| Chemical compound, drug | Neurobasal medium | GIBCO | A3582901 | |
| Chemical compound, drug | Neurocult SM1 neuronal supplement | STEMCELL | 05711 | |
| Chemical compound, drug | poly-lysine | Sigma Aldrich | P2636 | |
| Chemical compound, drug | 2-methylbutane | Sigma Aldrich | M32631 | |
| Chemical compound, drug | ethanol | Sigma Aldrich | 39278 | |
| Commercial assay, kit | ReadyProbes cell viability imaging kit Blue/Green | Thermo-Fischer Scientific | R37609 | |
| Software, algorithm | R. software | R. Core Team | v 3.6.1 | https://www.R-project.org/ |
| Software, algorithm | R-studio software | R-studio | v1.2.5001 | http://www.rstudio.com/ |
| Software, algorithm | ImageJ software | ImageJ | | http://imagej.nih.gov/ij/ |
| Software, algorithm | PyMCA software | PyMCA (ESRF) | | http://pymca.sourceforge.net/ |
| Other | SN, silicon nitride membranes | Silson Ltd, Southam, UK | SiRN-5.0 (o)−200–1.5-500 | Includes orientation frame |

## Culture of primary rat hippocampal neurons

Primary rat hippocampal neurons were cultured on silicon nitride (SN) membranes (Silson Ltd) consisting in square silicon frames of 5 × 5 mm² and 200 μm thickness with a central SN membrane of 1.5 × 1.5 mm² and 500 nm thickness. During manufacturing, a second, smaller (0.1 × 0.1 mm²), SN membrane is added in one of the corners of the silicon frame to serve as orientation object. Primary rat hippocampal neurons were dissociated from E18 Sprague-Dawley rat embryos (Janvier labs) and plated on the SN membranes previously treated with 1 mg.ml⁻¹ poly-lysine (Sigma Aldrich, P2636)

in a 0.1 M borate buffer pH 8.5. The SN membranes were placed on an astrocyte feeder layer growing on a dish treated by 0.1 mg.ml$^{-1}$ poly-lysine in a 0.1 M borate buffer pH 8.5, in Neurobasal medium (Gibco) supplemented with Neurocult SM1 neuronal supplement (STEMCELL) as adapted from the protocol of *Kaech and Banker, 2006*. At 3–4 days in vitro (DIV3-4), cell cultures were treated with 2 µM of cytosine arabinofuranoside (Sigma Aldrich) to limit the growth of the glial cells. From DIV6 and twice a week, half the Neurobasal medium was removed and replaced by BrainPhys medium (STEMCELL) at 310 mOsm, a culture medium designed to respect neuronal activity for in vitro models (*Bardy et al., 2015*). To develop dendritic spines, neurons were maintained in culture at 36.5°C in 5% $CO_2$ atmosphere until DIV15.

## Cell labeling for STED and confocal imaging

For live-cell microscopy of tubulin and actin, fluorogenic probes based on silicon rhodamine (SiR) from Spirochrome were used according to manufacturer instructions and published protocols (*Lukinavičius et al., 2014*; *Lukinavičius et al., 2016*). For single color STED imaging, SiR-tubulin or SiR-actin were added to DIV15 neurons in the BrainPhys medium at a final concentration of 1 µM during 1.5 hr at 37°C. For dual color imaging, SiR-tubulin and SiR700-actin were added to to the BrainPhys medium at 1 µM final concentration each and neurons were exposed during 1.5 hr at 37°C.

## STED and confocal imaging

Confocal and STED microscopy were performed on a commercial Leica DMI6000 TCS SP8 X microscope. DIV15 neurons cultured on SN membranes and labeled with SiR fluorogenic probes were maintained in the microscope chamber at 37°C in a Tyrode's solution (D-Glucose 1 mM, NaCl 135 mM, KCl 5 mM, $MgCl_2$ 0.4 mM, $CaCl_2$ 1.8 mM and HEPES 20 mM) pH 7.4 at 310 mOsm, the osmolarity of the BrainPhys culture medium. For live-cell microscopy the SN membrane was mounted in a Ludin chamber. The SN membrane is placed on the glass coverslip of the Ludin chamber, with neurons facing the coverslip to minimize the distance between the objective and the cells to be observed. Confocal and STED images were acquired with a HC-PL-APO-CS2 93x immersion objective in glycerol with a numerical aperture of 1.3 and a scan speed of 400 Hz. SiR fluorogenic probes were excited at 640 nm (670 nm for SiR700) and the signal was detected with a Leica HyD hybrid detector with a window of emission recording from 651 to 680 nm for SiR and from 710 to 750 for SiR700. For STED acquisitions, the fluorescence outside the center was quenched with a 775 nm pulsed diode laser synchronized with excitation.

## STED spatial resolution

The confocal lateral resolution was calculated as $\Delta d = \frac{0.4\lambda}{N.A}$ where $\Delta d$ is the smallest resolvable distance between two objects and N.A. is the numerical aperture. For confocal microscopy, the lateral resolution calculated was 194 nm. The STED lateral resolution was calculated as $\Delta d = \frac{0.5\lambda}{N.A\sqrt{1+\frac{1}{I_{SAT}}}}$. with $\lambda$ the emission wavelength and $\frac{1}{I_{SAT}}$ the saturation factor where I is the peak intensity of the depletion beam and $\Delta d = \frac{0.4\lambda}{N.A}$ the saturation intensity corresponding to the value of half emission signal. With a $\lambda$ emission wavelength for SiR-tubulin of 674 nm, a saturation factor $\frac{1}{I_{SAT}}$ between 34.5 and 57.5, and a N.A. of 1.3, the smallest distance between two objects that could be resolved was included between 32 and 44 nm. To fully exploit the spatial resolution of the STED setup (between 32 and 44 nm), the pixel size of STED images was set to 25 nm for oversampling the data.

## Plunge-freezing and freeze-drying

Immediately after STED observations the neurons were plunge-frozen. Samples were quickly rinsed in a 310 mOsm ammonium acetate solution, pH 7.4 to remove extracellular inorganic elements present in Tyrode's solution that would interfere with nano-SXRF element mapping. The osmolarity of Tyrode's and ammonium acetate solutions were measured with a vapor pressure osmometer (VAPRO 5600, Elite) and adjusted to the initial values of the BrainPhys culture medium (310 mOsm). Then the cells were blotted with Whatman paper. To avoid primary specimen distortions, the blotting is performed without touching the cells, by capillarity from the back and from the side frame of the SN membranes. Samples are plunge-frozen during 20 s in 2-methylbutane (Sigma Aldrich,

M32631) cooled down at −165℃ in liquid nitrogen. Excess 2-methylbutane was carefully blotted with Whatman paper cooled in liquid nitrogen vapors and samples transferred in the freeze-drier. Neurons were freeze-dried under gentle, minimally invasive conditions, during 2 days at −90℃ and 0.040 mbar in a Christ Alpha 1–4 freeze drier, allowing preservation of cell structure as verified in our previous work (*Perrin et al., 2015*). Then the temperature and the pressure were slowly raised up to room temperature and ambient pressure and the samples were stored at room temperature within a desiccator until synchrotron analyzes.

## Synchrotron nano X-ray Fluorescence microscopy (SXRF) and phase contrast imaging

Synchrotron experiments were performed on the ID16A Nano-Imaging beamline at the European Synchrotron Radiation Facility (Grenoble, France) (*da Silva et al., 2017*). The beamline is optimized for X-ray fluorescence imaging at 20 nm spatial resolution, as well as coherent hard X-ray imaging including in-line X-ray holography and X-ray ptychography. Featuring two pairs of multilayer coated Kirkpatrick-Baez (KB) focusing mirrors, the beamline provides a pink nanoprobe ($\Delta E/E \approx 1\%$) at two discrete energies: $E$ = 17 keV and 33.6 keV. Despite the larger focus at lower energy, for the correlative STED-SXRF experiment 17 keV was chosen as it is more efficient in exciting the X-ray fluorescence of the biologically relevant elements. The X-ray focus with dimensions of 35 nm (H) x 57 nm (V) provided a flux of 3.7 $10^{11}$ ph/s. The focus spot size was determined with a lithographic sample consisting of a 10 nm thick, $20 \times 20 \ \mu m^2$ square of nickel on a 500 nm thick SN membrane. The SN membranes holding the neurons were mounted in vacuum on a piezo nano-positioning stage with six short range actuators and regulated under the metrology of twelve capacitive sensors (*Villar et al., 2018*). An ultra-long working distance optical microscope was used to bring the sample to the focal plane (depth-of-focus±3 μm) and to position the STED regions of interest in the X-ray beam (see section sample positioning below). The samples were scanned with an isotropic pixel size of 40 nm, in some cases 20 nm for the scans of smaller size, and 100 ms of integration time. The X-ray fluorescence signal was detected with two custom energy-dispersive detectors (Rayspec Ltd) at either side of the sample, holding a total of ten silicon drift diodes. The quantitative data treatment of the SXRF data was performed with Python scripts exploiting the PyMCA library (*Solé et al., 2007*), using the fundamental parameter method with the equivalent detector surface determined by calibration with a thin film reference sample (AXO DRESDEN GmbH). The resulting elemental areal mass density maps (units: ng.mm$^{-2}$) were visualized with ImageJ. The X-ray phase contrast imaging (PCI) exploits in-line holography. It is performed on the same instrument moving the sample a few millimeters downstream of the focus and recording X-ray in-line holograms with a FReLoN CCD-based detector located 1.2 m downstream of the focus (*Mokso et al., 2007*). X-ray holograms were collected in the Fresnel region at four different focus-to-sample distances to assure efficient phase retrieval. At each distance, images at 17 different lateral sample positions were recorded and averaged after registration, to eliminate artefacts related to the coherent mixing of the incident wavefront and the object. The neurons being pure and weak phase objects, phase retrieval was performed using the contrast transfer function approach (*Cloetens et al., 1999*), implemented in ESRF inhouse code using the GNU Octave language. The phase maps, proportional to the projection of the electron density, had a final pixel size of 15 nm and a field of view of $30 \times 30 \ \mu m^2$. The spatial resolution was approximately 30 nm, similar to the STED and SXRF images.

## Sample positioning

During STED microscopy the orthonormal coordinates (x,y) of the regions of interest are recorded according to three reference positions on the SN frame, 3 corners of the square SN membrane (*Figure 1a*). The first reference position corresponds to the corner close to the 0.1 × 0.1 mm$^2$ orientation membrane, the second and third reference points to the corners selected from the first one in the clockwise direction. Using these three reference points the new coordinates (x′,y′) of the selected regions of interest can be calculated on the synchrotron ID16 setup using coordinate transformation equations (*Figure 1a*).

## Calculation of SXRF LOD

The limit of detection (LOD) obtained with the ID16A nano-SXRF setup was determined for the elements phosphorus, sulfur, potassium and zinc according to IUPAC guidelines (*McNaught and Wilkinson, 1997*).

$$LOD = m_{bi} + k \cdot \sigma_{bi} \tag{1}$$

where $m_{bi}$ is the mean of the blank measures, $\sigma_{bi}$ is the standard deviation of the blank measures, and k is a numerical factor chosen according to the confidence level desired, k = 3 for LOD. The resulting LOD values for each element based on the mean and standard deviation of 12 different blank analyses are presented in *Table 1*.

## Zinc chelation with TPEN

TPEN (Sigma Aldrich, P4413) cytotoxicity was controlled using the ReadyProbes cell viability imaging kit, Blue/Green, from Thermo-Fischer Scientific (R37609), according to the manufacturer instructions. Mature neurons (15 DIV) were treated with increasing concentrations of TPEN (0 to 10 µM), dissolved in analytical grade ethanol (Sigma Aldrich, 39278) to avoid trace element contamination, at a final ethanol concentration of 0.1% (v/v) in the culture medium, during 24 hr. Zinc and other chemical elements content was monitored in exposure solutions by PIXE (Particle Induced X-ray Emission) analysis. The counting of total and dead neurons was performed using fluorescence imaging on a Leica DM5000 microscope. The concentration of 5 µM TPEN was selected for further zinc chelation experiments since this concentration induced a subcytotoxic effect, with 83% of neurons alive compared to the control group (100%). Three experimental groups were compared by quantitative fluorescence microsocpy, neurons exposed to 5 µM TPEN for 24 hr, neurons exposed simultaneously to 5 µM TPEN and 10 µM $ZnCl_2$ for 24 hr, and control neurons exposed to the vehicle solution of 0.1% (v/v) ethanol in culture medium.

## Fluorescence labeling of tubulin and F-actin

Primary rat hippocampal neurons were cultured on glass coverslip on an astrocyte layer following the method developed by *Kaech and Banker, 2006*. Primary neurons were fixed 20 min at room temperature in 4% paraformaldehyde, 0.25% glutaraldehyde and 0.1% Triton-X100 in a cytoskeleton buffer with a final concentration of 10 mM MES (2-(N-morpholino)ethanesulfonic acid), 150 mM NaCl, 5 mM EGTA (ethylene glycol-bis(β-aminoethyl ether)-N,N,N′,N′-tetraacetic acid), 5 mM glucose and 5 mM $MgCl_2$ (pH 6,2). After rinsing in the same cytoskeleton buffer then in PBS (phosphate-buffered saline) the aldehyde fluorescence was quenched in 0.1% $NaBH_4$ in PBS during 5 min. After 5 min in 0.2% Triton-X100 the neurons were rinced in PBS and then blocked in 2% BSA in PBS during 45 min. Neurons were incubated 45 min with the primary antibody Mouse IgG1 anti-beta-tubulin (Sigma Aldrich, T4026) diluted in 2% BSA (1:3000 v/v) and blocked again 15 min in the 2% BSA solution. Neurons were incubated 30 min in darkness with phalloidin-Alexa647 (ThermoFisher Scientific, A22287) (1:40; v/v) and Goat anti Mouse IgG1 CF568 (Ozyme, BTM20248) (1:500, v/v) to label respectively F-actin and tubulin. Coverslips were rinced in PBS and neurons were fixed again in 2% paraformaldehyde in PBS for 10 min to stabilize the phalloidin before to be treated with 50 mM $NH_4Cl$ to quench aldehyde fluorescence.

## Quantitative fluorescence microscopy and data analysis

Microscopy images of F-actin and β-tubulin of neurons were collected the day after chemical fixation with a Leica DM5000 upright fluorescence microscope at 63x magnification. All fluorescence images were acquired on the same day using the same experimental conditions. For each experimental group (control, 5 µM TPEN and 5 µM TPEN + 10 µM Zn), between 2 and 4 coverslips were analyzed. For each coverslip an average of 20 images were recorded randomly. The experiments were repeated three times on neurons dissected from three different animals to ensure biological reproducibility. Data were treated using ImageJ software (http://imagej.nih.gov/ij/). F-actin and β-tubulin segmentation of the images was performed using histogram thresholding. The threshold value was defined as the value immediately greater than the bin value with the maximum counts in the histogram. Neuronal somas were removed from the images using clear selection to select only dendrite areas. Mean fluorescence intensities (counts/pixel) were calculated for each channel (F-actin and β-

tubulin) on each segmented image. Mean fluorescence intensities values per image of 70 µm x 100 µm were then used to plot the F-actin and β-tubulin intensity signal for each of the three experimentals groups (control, 5 µM TPEN and 5 µM TPEN + 10 µM Zn). Fluorescence intensity was normalized against the median intensity of the corresponding control group as presented in *Figure 6* and *Figure 6—figure supplement 1*.

### Statistical analysis

For nano-SXRF data analysis of element distributions, Pearson's correlation coefficients were calculated using R software (*R Development Core Team, 2019*) with rcmdr package (*Fox and Bouchet-Valat, 2019*). Pearson's correlation coefficients were calculated after exclusion of the background pixels. Background threshold value was evaluated as the mean value of the background pixels plus three times the standard deviation of the mean. For fluorescence microscopy experiments, fluorescence intensity was normalized against the median of the control group for each experiment. Normality of distribution and homogeneity of variances were respectively checked with Shapiro-Wilk test and Bartlett's test. Comparison of normalized intensity between two independent groups was performed using t-test or Mann-Whitney test if normal distribution or homogeneity of variance were not met. ANOVA was used for comparison of three groups, with Tukey post-hoc test for pairwise comparisons. Kruskal-Wallis test was run when assumptions of normality or homogeneity of variances were not met. Significant Kruskal-Wallis test ($p<0.01$) was followed by Dunn's test with p-values adjusted for pairwise comparisons (Holm's method). Statistical analysis was made using Rstudio v 1.2.5001 (*Team R Studio, 2015*), R software v3.6.1 (*R Development Core Team, 2019*) with rcmdr (*Fox and Bouchet-Valat, 2019*), tidyverse (*Wickham et al., 2019*), ggpubr (https://rpkgs.datanovia.com/ggpubr/), rstatix (https://rpkgs.datanovia.com/rstatix/) packages. For details see *Supplementary file 2*.

## Acknowledgements

This project was supported by a doctoral fellowship from the University of Bordeaux (FD), a grant from Centre National de la Recherche Scientifique (CNRS) through the MITI interdisciplinary program (RO), a PEPS grant from CNRS and IDEX Bordeaux (RO and DC), ERC grant ADOS (339541) and DynSynMem (787340) to DC and support from the Regional Council Nouvelle Aquitaine. SXRF experiments were peformed at the European Synchrotron Radiation Facility (ESRF), Grenoble, France. We are grateful to ESRF staff for assistance in using ID16A beamline. STXM experiments were performed at SOLEIL synchrotron, Gif-sur-Yvette, France. We acknowledge SOLEIL for provision of synchrotron radiation facilities and we would like to thank HERMES staff and F Porcaro from CENBG, for their help during the experiments. We acknowledge P Mascalchi and C Poujol from Bordeaux Imaging Center, part of the France BioImaging national infrastructure, for support in microscopy; C Breillat and N Retailleau from the Cell Biology facility of IINS for neuronal cell culture.

## Additional information

### Funding

| Funder | Grant reference number | Author |
|---|---|---|
| Centre National de la Recherche Scientifique | MITI Interdisciplinary Program | Richard Ortega |
| European Research Council | DynSynMem (787340) | Daniel Choquet |
| IDEX Bordeaux | PEPS CorXsyn | Daniel Choquet Richard Ortega |
| University of Bordeaux | Doctoral Fellowship | Florelle Domart |
| Centre National de la Recherche Scientifique | PEPS CorXsyn | Daniel Choquet Richard Ortega |
| European Research Council | ADOS (339541) | Daniel Choquet |
| SOLEIL Synchrotron | beamtime allocation | Richard Ortega |

| | | |
|---|---|---|
| European Synchrotron Radiation Facility | beamtime allocation | Richard Ortega |
| Aquitaine Regional Council | | Daniel Choquet |

The funders had no role in study design, data collection and interpretation, or the decision to submit the work for publication.

### Author contributions

Florelle Domart, Conceptualization, Data curation, Formal analysis, Investigation, Methodology, Writing - original draft, Writing - review and editing; Peter Cloetens, Resources, Data curation, Software, Formal analysis, Methodology, Writing - review and editing; Stéphane Roudeau, Data curation, Formal analysis, Supervision, Investigation, Visualization, Methodology, Writing - review and editing; Asuncion Carmona, Data curation, Formal analysis, Supervision, Validation, Investigation, Methodology, Writing - original draft, Writing - review and editing; Emeline Verdier, Validation, Investigation, Methodology, Writing - review and editing; Daniel Choquet, Conceptualization, Resources, Formal analysis, Supervision, Funding acquisition, Validation, Investigation, Project administration, Writing - review and editing; Richard Ortega, Conceptualization, Resources, Data curation, Formal analysis, Supervision, Funding acquisition, Validation, Investigation, Visualization, Methodology, Writing - original draft, Project administration, Writing - review and editing

### Author ORCIDs

Florelle Domart (ID) https://orcid.org/0000-0001-5938-1731
Peter Cloetens (ID) https://orcid.org/0000-0002-4129-9091
Stéphane Roudeau (ID) https://orcid.org/0000-0002-4539-9380
Asuncion Carmona (ID) https://orcid.org/0000-0002-9253-4581
Daniel Choquet (ID) https://orcid.org/0000-0003-4726-9763
Richard Ortega (ID) https://orcid.org/0000-0003-1692-5406

### Decision letter and Author response

Decision letter https://doi.org/10.7554/eLife.62334.sa1
Author response https://doi.org/10.7554/eLife.62334.sa2

## Additional files

### Supplementary files

• Source data 1. Data source for *Supplementary file 1*.

• Supplementary file 1. Nano-SXRF quantitative data. Analysis of chemical elements content for 21 regions showing zinc and tubulin co-localization, expressed in $ng.mm^{-2}$ and in $atoms.nm^{-2}$ (mean ± SD, n = 21).

• Supplementary file 2. Statistical analysis of data.

• Transparent reporting form

### Data availability

Synchrotron datasets (SXRF and PCI images) are available from the ESRF data portal in open mode in two datasets. Data sets are associated to the following DOI numbers: doi:10.15151/ESRF-ES-162248067 and doi:10.15151/ESRF-ES-101127303. The link to the first dataset is https://data.esrf.fr/investigation/162248067/datasets, sign in as anonymous to access data. The link to the second dataset is https://data.esrf.fr/investigation/101127303/datasets, sign in as anonymous then click on the zoom button to reveal the download button. Source data for Figure 1 are available at https://data.esrf.fr/investigation/162248067/datasets; datasets M20_zone67_nfp3_015nm and M20_zone67_fine01. Source data for Table 1 are included in file Table 1—source data 1. Source data for Figure 2 are available at https://data.esrf.fr/investigation/101127303/datasets; datasets TA15_neu64_fine2 and TA15_neu64_fine5. Synchrotron XRF data for Figure 2—figure supplement 1 are available at https://data.esrf.fr/investigation/101127303/datasets; datasets TA15_neu64_fine4 and TA15_neu64_

fine3. Data for Pearson's correlation coefficients of Figure 2—figure supplement 1 panel h are provided in Figure 2—figure supplement 1—source data 1. Data for Pearson's correlation coefficients of Figure 2—figure supplement 1 panel o are provided in Figure 2—figure supplement 1—source data 2. Source data for Figure 3 are available at https://data.esrf.fr/investigation/162248067/datasets; datasets M8_neur43_sted44_nfp_015nm and M8_neu43_fine03. Synchrotron XRF data for Figure 3—figure supplement 1 are available at https://data.esrf.fr/investigation/101127303/datasets; dataset SiTA1_neu7_fine01. Source data for Figure 4 are available at https://data.esrf.fr/investigation/101127303/datasets; dataset TA15_neu71_fine01. Data for Pearson's correlation coefficients are included in file Figure 4—source data 1. Synchrotron XRF and PCI data for Figure 4—figure supplement 1 are available at https://data.esrf.fr/investigation/162248067/datasets; datasets M20_zone67_fine01, M20_zone67_fine02, and M20_zone67_fine06. Source data for Figure 5 are available at https://data.esrf.fr/investigation/101127303/datasets; datasets TA15_neu26_fine 01 and TA15_neu23_fine02. Synchrotron XRF data for Figure 5—figure supplement 1 are available at https://data.esrf.fr/investigation/162248067/datasets; datasets M20_zone67_nfp3_015nm and M20_zone67_fine01. F-actin data for Figure 6 are available in file Figure 6—source data 1. β-tubulin data for Figure 6 are available in file Figure 6—source data 2. F-actin data for Figure 6-figure supplement 1 are available in file Figure 6—figure supplement 1—source data 1. Tubulin data for Figure 6-figure supplement 1 are available in file Figure 6—figure supplement 1—source data 2. Supplementary file 1 raw data provided in Source data 1.

The following datasets were generated:

| Author(s) | Year | Dataset title | Dataset URL | Database and Identifier |
|---|---|---|---|---|
| Ortega R, Roudeau S, Domart F | 2018 | Correlative super resolution SXRF and STED imaging of biological metals and synaptic proteins in frozen hydrated hippocampal neurons. European Synchrotron Radiation Facility (ESRF) | https://data.esrf.fr/investigation/162248067/datasets | ESRF data portal, 10.15151/ESRF-ES-162248067 |
| Cloetens P, Yang Y, Ortega R, Domart F | 2018 | Correlative X-ray microscopy and super-resolution microscopy of freeze dried hippocampal neurons. European Synchrotron Radiation Facility (ESRF) | https://data.esrf.fr/investigation/101127303/datasets | ESRF data portal, 10.15151/ESRF-ES-101127303 |

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
