## [Decision Letter]

**Acceptance summary:**

The manuscript clearly presents an innovative and novel technique, which will potentially provide new biological insights into metal distribution in cells.

**Decision letter after peer review:**

Thank you for submitting your article "Correlating STED and synchrotron XRF nano-imaging unveils cosegregation of metals and cytoskeleton proteins in dendrites" for consideration by *eLife*. Your article has been reviewed by two peer reviewers, and the evaluation has been overseen by John Kuriyan as the Reviewing Editor and Senior Editor. The reviewers have opted to remain anonymous.

The reviewers have discussed the reviews with one another and the Reviewing Editor has drafted this decision to help you prepare a revised submission.

In this manuscript, the authors developed a new technique combining STED super-resolution fluorescence imaging and synchrotron X-ray fluorescence nano-imaging (SXRF). SXRF imaging allows them to image metals, such as Zn and Cu, in neurons. They demonstrate super-resolution images of the localization of metals by SXRF together with the cytoskeleton with STED. In particular, they conclude that Zn is colocalized with microtubules, likely as a protein complex, in dendrites. By correlating with the number of known elements (like S and P) in microtubules, they infer that ~1 Zn per the α-β pair of tubulin molecules. The neural cytoskeleton was reduced by chelating Zn2+, further suggesting the important roles of Zn. The authors also observed colocalization of Cu and F-actin in F-actin-rich structures and conclude that Cu either binds to F-actin or to other biomolecules that interact with F-actin.

Overall, the manuscript clearly demonstrates a novel and innovative microscopy technique that potentially provide many insights into the metal distribution and new biology associated with it. The experimental work is solid. The images derived from the correlative microscopy approaches point to a herculean effort of combining two labor intensive methods. Sample preparation, data collection, analysis and presented images are of highest quality. The co-localization of metals and proteins are clearly shown.

Reviewers have expressed one major reservation about this work, which is the rigor with which the colocalization data, which are just correlations, are interpreted in terms of the actual molecular events that lead to the observed colocalization. We note that this paper is being considered for the Tools and Resources section of *eLife*, and as such the emphasis should be on the methods, and less on the mechanistic interpretation. With this in mind, please respond to the two major issues below by either rewording the text to highlight limitations in interpretation, or provide new experimental data that support the conclusions. You should also respond to all of the other comments in a revised manuscript.

Essential revisions:

1) The authors find that Zn and microtubules (tubulin) are colocalized with Zn and conclude that Zn is binding to tubulin as a structural component. They substantiate the claim by citing >15 year old proteomics data for HepG2 and Hela cells that used IMAC to identify metal-binding proteins in cells. They also use the colocalization of S and Zn as further evidence that Zn binds to tubulin in a 1:1 ratio based on the number of S in each protein. These are at best correlations and should have been followed up by other experiments. Instead the authors cultured neurons under Zn depleted conditions and show a significant decrease in tubulin and F-actin. They discuss these observations at length but it is not clear whether they are suggest that this further supports their hypothesis of Zn binding to tubulin.

2) Similarly, Cu co-localizes with F-actin but not everywhere for which the authors offer two theories: either the sensitivity is too low, or Cu does not directly bind to F-actin. Again, the colocalization of Cu and F-actin would have been a more impactful conclusion had this observation been followed up by experiments to more directly demonstrate the nature of the interaction.

---

## [Author Response]

Essential revisions:1) The authors find that Zn and microtubules (tubulin) are colocalized with Zn and conclude that Zn is binding to tubulin as a structural component. They substantiate the claim by citing >15 year old proteomics data for HepG2 and Hela cells that used IMAC to identify metal-binding proteins in cells. They also use the colocalization of S and Zn as further evidence that Zn binds to tubulin in a 1:1 ratio based on the number of S in each protein. These are at best correlations and should have been followed up by other experiments. Instead the authors cultured neurons under Zn depleted conditions and show a significant decrease in tubulin and F-actin. They discuss these observations at length but it is not clear whether they are suggest that this further supports their hypothesis of Zn binding to tubulin.

The discussion of this part was probably too extended and the conclusions too categorical. Experiments are in progress to confirm Zn binding to tubulin, and also the binding of copper to Factin (see comment #2). However, results will not be ready for publication probably before several months so we cannot present them for the revision of this article. As suggested, we have modified the text to focus more on the methodological development than on the biological results:

– The discussion now begins with the text related to the methods.

– The Discussion section about Zn localization in spines is now placed before the discussion of Zn co-localization with tubulin to make it more visible and to mirror the organization of the Results section.

– The discussion about Zn interaction with tubulin has been shortened from initially 793 words to 371 words.

– In particular, the text referring to the IMAC data was removed, it was not essential, we only kept the references on molecular modeling, biophysical and biochemical evidence. Part of the discussion on Zn/tubulin quantitative ratio calculation was redundant with the Results section and was simplified.

– A large part of the discussion on Zn chelation results and its effects on tubulin expression has been removed.

– The discussion text on Zn interaction with tubulin was written in a less categorical way.

– The last paragraph of the general Discussion has been fully modified to insist less on the biological conclusions and to emphasize the need for further experiments.

2) Similarly, Cu co-localizes with F-actin but not everywhere for which the authors offer two theories: either the sensitivity is too low, or Cu does not directly bind to F-actin. Again, the colocalization of Cu and F-actin would have been a more impactful conclusion had this observation been followed up by experiments to more directly demonstrate the nature of the interaction.

These experiments are under progress. We have already obtained some results proving the direct binding of Cu to cysteine residues from purified (commercial) F-actin using mass spectrometry. However to show the direct binding of Cu to F-actin extracted from a complex protein mixture is much more challenging (without denaturation and remetallation of the protein). We are currently working on these protocols (same for Zn-tubulin binding). Since we will not be able to add new results to this revised manuscript and to answer to the reviewer and editor suggestions we have modified the text as follows:

– The discussion about Cu interaction with F-actin has been shortened from initially 451 words to 275 words to give less emphasis on this result.

– Most of the text referring to the IMAC data on Cu was removed, it was too long.

– The discussion text on Cu interaction with F-actin was written in a less categorical way.

– The last paragraph of the general Discussion was fully modified to insist less on the biological conclusions and to emphasize the requirement for further experiments.